# Genomic and Chemical Profiling of B9, a Unique *Penicillium* Fungus Derived from Sponge

**DOI:** 10.3390/jof8070686

**Published:** 2022-06-29

**Authors:** Chaoyi Chen, Jiangfeng Qi, Yajing He, Yuanyuan Lu, Ying Wang

**Affiliations:** School of Life Science and Technology, China Pharmaceutical University, Nanjing 211198, China; 13770998012@163.com (C.C.); qijiangfenglalala@163.com (J.Q.); 3320031562@stu.cpu.edu.cn (Y.H.)

**Keywords:** *Penicillium* sp., genome sequence, BGCs, secondary metabolites

## Abstract

This study presented the first insights into the genomic and chemical profiles of B9, a specific Penicillium strain derived from sponges of the South China Sea that demonstrated the closest morphological and phylogenetic affinity to *P. paxillin*. Via the Illumina MiSeq sequencing platform, the draft genome was sequenced, along with structural assembly and functional annotation. There were 34 biosynthetic gene clusters (BGCs) predicted against the antiSMASH database, but only 4 gene clusters could be allocated to known BGCs (≥50% identities). Meanwhile, the comparison between B9 and *P. paxillin* ATCC 10480 demonstrated clear distinctions in morphology, which might be ascribed to the unique environmental adaptability of marine endosymbionts. In addition, two novel pyridinones, penicidihydropyridone A (2) and penicidihydropyridone B (3), were isolated from cultures of B9, and structurally characterized by nuclear magnetic resonance (NMR) and mass spectrometry (MS). The absolute configurations were confirmed by comparison of experimental and calculated electronic circular dichroism (ECD) curves. In addition, structure-based molecular docking indicated that both neo-pyridinones might block the programmed cell death protein 1(PD-1) pathway by competitively binding a programmed cell death 1 ligand 1(PD-L1) dimer. This was verified by the significant inhibition rates of the PD-1/L1 interaction. These indicated that *Penicillium* sp. B9 possessed a potential source of active secondary metabolites.

## 1. Introduction

The ocean covers more than 70% of the Earth’s surface and hosts huge biological diversity, the vast majority of which remains unexplored. Of note, marine-derived microorganisms, due to their enormous biodiversity and robust environmental adaptability, present a tremendous reservoir of valuable natural products and metabolic intermediates. Characterized by structural novelty and functional complexity, they are potentially applicable in industrial, agricultural, food, cosmetic, and pharmaceutical fields [1].

Marine microeukaryotes, fungi, and particularly, *Penicillium* species, have attracted recent interest in the field of microbial biodiversity and ecology, due to their secondary metabolites (SMs), unusual chemical architectures, and alluring biological features [2]. *P. paxilli*, a saprophytic anamorph of the genus *Penicillium*, is used as a model organism to study the biology of indol-diterpene production, with the intriguing biosynthetic potential of terpenomes, e.g., paxilline, paxisterol, penicillone, pyrenocine A, paspaline B, etc. [3]. The indole diterpenoid compounds comprise an indole moiety fused to a diterpene skeleton, many of which act as bioprotectants. This has the evolutionary advantage of offering protection from predators by causing neurological disorders in farm animals [4,5]. For example, paxilline is a tremorgenic mycotoxin that arises from big potassium (BK) channel inhibition [6], which has been consequently applied in various biophysical and physiologically relevant studies [7]. In a previous study from our group, a series of filamentous fungal strains, predominantly *Penicillium* and *Aspergillus sp*., have been identified from sponges in the South China Sea [8]. Among them, a putative subspecies of *P. paxilli* was defined in this study by morphological and phylogenetic analysis, which underlined the variations that might ascribe to the adaptative evolution to niche environments. Hence, the draft genome of the strain was sequenced, for which, genomic mining and functional prediction were subsequently performed. Meanwhile, chemical separation and structural characterization of the SMs presented two novel compounds, penicipyridinone A (2) and penicipyridinone B (3), which intriguingly, appeared to perturb PD-L1/PD-1 interactions in vitro. Demonstrably, these findings provided some insight into phylogeographic flexibility and physiological and biochemical diversity of marine-derived endosymbiotic *Penicillium* sp.

## 2. Materials and Methods

### 2.1. Stains

The B9 strain was isolated from sponges in reefs of the South China Sea (17°5′0.70″ N, 111°31′1.39″ E). Followed by consecutive purification, the strain was inoculated into potato dextrose agar (PDA; 20 % potato, 2% dextrose, 1.5% agar, and natural pH) at 28 ℃ for 3 days to obtain colonies for species identification or culture preservation.

The reference strain in this study, *P. paxillin* ATCC 10480, acquired from the Guangdong Microbial Culture Collection Center (GDMCC, China), was cultured and maintained under the same conditions as B9.

### 2.2. Species Identification

For the purified colony, preliminary identification was conducted by morphological analysis as described [9]. Briefly, the spore suspensions of B9 were three-point inoculated onto malt extract agar (MEA; OXOID Ltd., Basingstoke, UK) with 3% sea salt, and cultured at 28 °C for 5 days. Colony morphologies were assessed in terms of shape, size, texture, and nature of colony, specifically mycelium production, substratum mycelium (pigmentation), and aerial mycelium (spores). Microscopic examination was then performed on hyphae and conidiophore structure, followed by lactophenol cotton blue (LPCB) staining [10] under a BA210 light microscope (Motic, Xiamen, Fujian, China).

Molecular identification was implemented by sequencing analysis of the fungal ITS sequence. Genomic DNA was prepared for gene amplification from mycelium of 5-day culture on MEA. The ITS region was amplified and sequenced by primers ITS1 and ITS4 (ITS1,5′-TCCGTAGGTGAACCTGCGG-3′; ITS4, 5′-TCC TCCGCTTATTGATATGC-3′) [11]. The full-length ITS sequence was analyzed through the Basic Local Alignment Search Tool (BLAST), and aligned with homologous ITS sequences by the ClustalW multiple alignment program. A neighbor-joining phylogenetic tree was then generated with 1000 bootstrap replicates in MEGA7 [12].

### 2.3. Intraspecies Comparison

In a phylogenetic context, the B9 strain was placed into a sister branch of *P. paxillin* in *penicillium* sp., supposedly as a close relative of adaptative stress. Hence, a terrigenous reference strain, *P. paxillin* ATCC 10480, was hired for physiological and morphological analogy. Both strains were grown in identical conditions. In brief, after 3 days’ activation at 28 °C on MEA, the fungal spores were resuspended in sterile phosphate-buffered saline (PBS) adjusted to 0.5 × 10^7^ spores/mL. 3 µL aliquots of spore suspension were inoculated onto MEA plates. Macro and microscopic morphological features were observed on the 5th day of culture at 28 °C.

### 2.4. Whole Genome Sequencing

Whole genomes of B9 were sequenced on an Illumina Miseq platform by Genewiz Co. (Suzhou, China) as described [8]. In brief, sequencing libraries were constructed from 100 ng genomic DNA samples following the standard Illumina protocol. Sequencing was performed on HiSeq Control Software in the paired-end (2 × 150 bp) mode. Adapter sequences were removed, and low-quality reads were trimmed by Cutadapt (v1.9.1). De novo assemblies were then generated using an iterative process involving Velvet, SSPACE, and GapFiller. The sequencing data were deposited in the NCBI SRA database with accession PRJNA808833.

### 2.5. Functional Genome Annotation

The draft genome was annotated by Augustus gene prediction software (version 3.3) [13], with *P. paxilli* ATCC 26601 as a reference. The assembled genes were BLAST-searched against the NCBI nr database and functionally annotated into the KOG [14] and KEGG [15] databases. Finally, genomic regions containing SM BGCs and putative SMs were predicted on the antiSMASH [16] server with the ClusterFinder algorithm.

### 2.6. Extractions of Secondary Metabolites (SMs)

Following activation for 3 days at 28 °C, a spore suspension of B9 was obtained and used as the inoculum. 10 mL of spores (1 × 10^7^ spores/mL) were then inoculated on rice medium (rice 200 g, sea salt 3 g, MgSO_4_ 0.2 g, H_2_O 200 mL) and cultured at 28 °C for 100 days. The fermented products from 8 kg of rice medium were extracted with 2.5 volumes of methanol (MeOH) for 2 h and filtered. The methanol extract was subjected to liquid–liquid extraction with ethyl acetate (EtAc) and concentrated by reduced pressure distillation. The crude product (45 g) was purified via silica gel column chromatography with a stepwise gradient of CHCl_3_/MeOH (100:0, 50:1, 40: 1, 30:1, 20:1, 10:1, 5:1, 2:1, 1:1, and 1:2; vol/vol), to give ten fractions (fractions A–J). Fraction D (800 mg) was further purified by silica gel (200–300 mesh, Qingdao Marine Chemical Inc., Qingdao, Shandong, China) and ODS (50 μm; YMC Co., Ltd., Komatsu, Japan) chromatography (H_2_O-MeOH gradient system). Final purifications of compounds 1–3 were achieved by preparative C18 High-performance Liquid Chromatography (HPLC) using an Rp-C18 column (5 μm, 10 × 250 mm; Hanbon, Huan’an, China) with MeOH/H_2_O (18:82) at 3.5 mL/min and detection at 254 nm to yield 300, 60, and 10 mg of purified compounds, respectively.

Penicipyridinone A: white powder; ${[\alpha ]\;}_{D}^{20}$-8.0 (c 0.05, EtOH); UV (MeOH) λmax (log ε): 303 (3.18) and 264 (3.08) nm; IR ν_max_: 3248, 3069, 1656, 1601, 1514, 1412, 1175, 747, 690, 604, 521 cm^−1^; ^1^H NMR and ^13^C NMR, see Table 1; HR-ESI-MS m/z 184.0962 [M+H] ^+^, (calculated for C_9_H_13_NO_3_, 184.0968).

Penicipyridinone B: colorless gum; ${[\alpha ]\;}_{D}^{20}$-8.0 (c 0.05, EtOH); UV (MeOH) λmax (log ε): 272 (2.98), 247 (3.08) and 204 (2.81) nm; IR ν_max_: 3353, 3253, 2968, 1605, 1506, 1320, 854, 663, 597 cm^−1^; ^1^H NMR and ^13^C NMR, see Table 2; HR-ESI-MS m/z 200.0925 [M+H] ^+^, (calculated for C_9_H_13_NO_4_, 200.0917).

### 2.7. In Vitro PD-1/PD-L1 Binding Assay

A competitive PD-1/PD-L1 binding assay was designed to evaluate the inhibitory potentials of compounds on PD-1/PD-L1 interaction by a homogeneous time-resolved fluorescence (HTRF)-based kit (Cisbio, Bedford, MA, USA). Briefly, 20 ng compound, Tag1-PD-L1 protein (100 nM), and Tag2-PD-1 (100 nM) were added into a 384-well plate successively and then incubated at room temperature for 15 min. 10 μL of pre-mixed detection reagent containing anti-Tag1-Europium (HTRF donor) and anti-Tag2-XL665 (HTRF acceptor) was added to measure the fluorescence energy transfer between donor and acceptor. After 2 h of incubation, fluorescence signals were obtained on a Spark plate reader (Tecan, Männedorf, Switzerland). The ratios of XL665 emissions at 665 nm to Eu emissions at 620 nm were measured to derive the HTRF signal of each compound. The binding inhibition rate was calculated (combined control signal–compound signal)/(combined control signal–background signal).

### 2.8. Docking Studies

A molecular docking study was performed using a Discovery Studio 4.5. CDOCKER based on a CHARMm-based MD docking algorithm, which was used to accurately dock the designed ligands into the PD-L1 dimer. A crystal structure of the PD-L1/inhibitor complex at a high-resolution of 1.7 Å was selected as the docking target (PDB code 5N2F). The binding site of the crystal ligand was defined as the docking site. The reproduction of the crystal pose was first carried out. The default settings were applied during the docking studies. The top 10 hits were ranked according to their CDOCKER energies (interaction energy plus ligand strain), and the top-scoring (most negative, thus favorable to binding) poses were selected for the further binding mode analysis.

## 3. Results

### 3.1. Species Identification of Strain B9

Based on the culture, isolation, and morphological observation, strain B9 was initially classified into the genus *Penicillium*. The colonies grew vigorously on MEA and attained a diameter of 22 mm after 5 days of incubation at 28 °C, spreading with aerial mycelium and smooth, regular margins. The front color of colonies was teal with a pale white peripheral border, and yellowish to dark brown in reverse, with velutinous to floccose texture. Under the optical microscope, mycelium was separate filiform, with 3–5 μm spherical conidia clustering on inflated phialides to form broom-shaped hyphae, typically demonstrating the morphologic characteristics of *Penicillium* sp. (Figure 1A).

With the fungal-specific primers (ITS1/ITS4), the full-length internal transcribed spacer (ITS) region was amplified and sequenced. Following multiple sequence alignment analyses, a maximum-likelihood phylogenetic tree was generated. As shown in Figure 1B, the B9 strain was allocated into a separate branch closely associated with the *P. paxilli* clade with the highest identities (over 98%). Within the *Penicillium* genus, the subcluster was more distant than other species, supported by the similarity values ranging from 89% to 96%. In terms of phylogeny, the strain could be a distinct species of *Penicillium*, as a close relative of *P. paxilli*, which certainly requires further investigation.

Subsequently, a type strain of *P. paxilli*, ATCC 10480, was hired for interspecies comparison of phenotypic and ecological adaptations and reflected genetic affinity to some extent. Macroscopically and microscopically, the morphology of B9 closely resembled that of the type strain, except for a darker reverse of the colony, probably hinting at the increased production of liposoluble pigments (Figure 1A). As shown in Figure 1C, both strains grew well at a series of salinity (0–7%) and pH (4–9) levels, demonstrating the broad tolerances of *Penicillium* sp. In terms of pigmentation, the B9 strain was more resilient to pH range than salinity alteration, with robust fitness in alkalinity (pH = 8–9). Seemingly, B9 might be an exceptional strain in Penicillium sp., represented by the different ecophysiological characteristics, even from its closest phylogenetic relative *P. paxilli*, for which, further investigation was necessary.

### 3.2. Genome Sequencing

To derive further genetic information, the whole genome of B9 was sequenced with a coverage of 189.27×. The draft genome was assembled into a total size of 32.96 Mb, with a G+C content of 47.7%, comprising 185 scaffolds. The average length of consensus contigs was 186,810 bp with an N50 of 1,131,010 bp. A total of 11,110 protein-coding genes were predicted, characterized by an average gene length of 1545.9 bp. The general characteristics of the B9 genome are listed in Table 1.

### 3.3. Genomic Functional Annotation

Based on the primary structure homology, the B9 genome was functionally annotated in the Cluster of Orthologous Groups of proteins (KOG) and the Kyoto Encyclopedia of Genes and Genomes (KEGG) databases. In the KOG classification, 6352 putative proteins were assigned, accounting for 57.17% of the total protein-coding sequences (Figure 2), which were categorized into 4 main groups: intracellular processes (20.38%), metabolism (37.89%), information storage processing (18.23%), and poorly characterized function (23.50%). Among the assumed pathways, metabolism appeared as the most prominent functional category, including amino acid metabolism (469 genes), lipid metabolism (433 genes), carbohydrate metabolism (421 genes), and secondary metabolism (415 genes). Aside from the common pathways of primary metabolism, secondary metabolic pathways were enriched in the genome of B9, likely representing the great potential for the biosynthesis of SMs.

Moreover, KEGG annotation consistently suggested that the encoding genes were enriched during metabolic processes, with secondary metabolic functions being specifically distributed to the biosynthesis of terpenoids, polyketones, and other metabolites (Figure 3). Among the allocated 46 KEGG subcategories, signal transduction was also significantly represented, which might be implicated as a molecular basis for the environmental plasticity of the strain.

### 3.4. Prediction of the Biosynthetic Gene Clusters (BGCs)

In fungi, genes involved in secondary metabolite biosynthesis are often spatially clustered as BGCs [17]. Hence, the potential BGCs in the B9 genome were predicted in the antibiotics & Secondary Metabolite Analysis Shell (antiSMASH) database, which identified 34 putative BGCs, composed of 20 non-ribosomal peptide synthases (NRPSs), 9 polyketide synthases (PKSs), 2 terpenes, 2 NRPS-indole hybrids, and 1 beta-lactone cluster (Figure 4). Compared with the reference genome of Penicillium paxilli ATCC 26601, BGCs demonstrated more variation in B9, whether measured as a number or as diversity (Figure 4). Among the heterogeneous classes, the multidomain mega-enzymes, NRPSs, were the most abundant (>50%). They are commonly known to be involved in the assembly line-like synthesis of numerous peptide natural products associated with many well-known pharmacophores [18]. In addition, PKSs were plenteous and were the next abundant type in the B9 genome. This is consistent with the KEGG functional analysis, collectively implying the robust potentials of secondary metabolism.

Further analysis of the BGCs revealed that most of the hypothetical proteins or domains were uncharacterized with unknown functions or products. More specifically, there were 8 BGCs with variable degrees of similarity to known BGCs, corresponding to different synthetic profiles such as NRPS (nidulanin A, Dimethylcoprogen) (Figure 5A,B), PKS (Naphthopyrone) (Figure 5C), terpene (squalestatin S1) (Figure 5D), NRPS-indole hybrid, etc. The putative products come in various sizes and constructions, with functions ranging from extracellular siderophores (dimethylcoprogen) [19], acetylcholine esterase (AChE) inhibitor (naphthopyrone) [20], squalene synthase inhibitor (squalestatin S1) [21] and phytotoxins (solanapyrone) [22].

As for the large proportion of BGCs with unknown functions present in the genome, further analyses were conducted. For example, the presumptive BGC (*pax*) of paxilline, one of the signature metabolites of *P. paxillin*, displayed significant homology of gene architecture and sequence identity to the reported clusters (BGC0001082) or to the orthologue in the reference genome (GCA_000347475) (Figure 6). However, it could not be annotated in the antiSMASH database. The case should largely ascribe to the limited information on the structural and functional properties of the biosynthetic pathways available in the database, which is likely insufficient to support a comprehensive assessment of the strain based solely on the genome-mining scope.

### 3.5. SMs Isolation and Chemical Characterization

The strain B9 was fermented for SM extraction by chemical separation and characterization, offering paxilline (1) and two novel compounds (Figure 7): penicidihydropyridone A (2) and penicidihydropyridone B (3). From NMR and elemental analyses, the *structural elucidation* of both new compounds was studied.

Compound **2** gave a pseudo molecular ion [M+H] ^+^ peak at m/z 184.0962 (Calculated for m/z 184.0968, Δ 3 ppm) by high-resolution electrospray ionization mass spectroscopy (HR-ESI-MS), consistent with the molecular formula C_9_H_13_NO_3_ (Appendix A). There are two single methyls (δ_H_ 2.41 (3H, s, H-8) and 2.37 (3H, s, H-9)), one doublet methyl group (δ_H_ 1.32 (3H, d, H-10)), one methylene group (δ_H_ 2.49 (1H, dd, H-5a) and 2.34(1H, dd, H-5b)), and one methane group (δ_H_ 3.76(1H, m, H-6)) presenting in the ^1^H NMR spectroscopy, respectively (Table 2; Appendix A). In ^13^C NMR spectroscopy, 9 carbon signals were detected, in which four carbon signals, δ_C_ 109.7 (C-3), 170.7 (C-2), 191.8 (C-4), and 198.9 (C-7), revealed structural features of compound **2** nearly identical to those of 3-acyltetronic acids (Table 2; Appendix A). In the Heteronuclear Multiple Bond Correlation (HMBC) spectrum, the correlation peak between δ_H_ 3.76 (H-6) and δ_C_ 170.7 (C-2) indicates that compound **2** might be a lactam skeleton type structure (Appendix A). Similarly, the cross-peaks between H-9 (δ_H_ 2.37) and C-2, and H-5 (δ_H_ 2.34) and C-4 (δ_C_ 191.8), lend strong support to the elucidation structure of compound **2** (Table 2). Furthermore, the stereochemical structure of carbon 6 was identified as S according to ECD methods (Figure 8; Appendix A). Thus, compound **2** was elucidated as (6S)-3-acetyl-4-hydroxy-1,6-dimethyl-5,6-dihydropyridin-2(1H)-one. It is given the trivial name, penicidihydropyridone A.

Compound **3** was obtained as a pseudo molecular ion [M+H] ^+^ peak at m/z 200.0925 (Calculated for m/z 200.0917, Δ 4 ppm) by HRESIMS, consistent with the molecular formula C_9_H_13_NO_4_(Table 3; Appendix A). In the ^1^H NMR spectrum, compounds have one ethyl group (δ_H_ 1.66 (1H, dt, J = 17.5, 6.2 Hz, H9a), 1.73 (1H, dt, J = 17.5, 6.2 Hz, H9b) and 0.89 (3H, t, J = 6.2 Hz, H10)), one methylene group (δ_H_ 2.87 (1H, d, J = 15.2, H-5a) and 2.34 (1H, dd, J = 15.2, H-5b)), and one methyl group (δ_H_ 2.42 (3H, s, H-8)) (Table 3; Appendix A). The ^13^C NMR also afforded 9 carbons, including δ_C_ 109.7 (C-3), 179.7 (C-2), 201.6 (C-4), and 197.1 (C-7) (Table 3; Appendix A). From NMR data of compound **3**, it has a similar 3-acyltetronic acid fragment as compared to 2. In the HMBC spectrum, the correlation peak between δ_H_ 6.85 (H-10), δ_C_ 77.3 (C-6), and 30.1 (C-9) indicates that the ethyl group was substituted at C-6 (Appendix A). The absolute configuration of C-6 was determined by the DP4^+^ calculations. As shown in Appendix A, 6R displayed a slightly better fit between experimental and calculated ^13^C NMR data and was identified as the most probable candidate. Thus, the structure of compound **3** was identified as (6R)-3-acetyl-6-ethyl-4,6-dihydroxy-5,6-dihydropyridin-2(1H)-one. It is given the trivial name, penicidihydropyridone B.

### 3.6. Biological Activities of New Compounds

The docking simulation of compounds **2** and **3** to the PD-L1 dimer (PDB:ID 5N2F) (Figure 9) resulted in the interaction energies. After the CHARMm energy minimization, the total interaction energy for the complex was equal to −5.6 kcal/mol for compound **2** and −6.1 kcal/mol for compound **3**, which supposedly implicated the antagonistic effects on the PD-1 pathway.

Based on the Homogeneous Time-Resolved Fluorescence (HTRF) assay system, blocking potencies of the new compounds (**2** & **3**) on the PD-1 axis were evaluated. The bioactivity assessment validated the calculation analysis, demonstrated by the considerable inhibitory rate of PD-1/PD-L1 interactions at 88.40% for compound **2** and 70.72% for compound **3**, at the concentration of 10 μg/mL, highlighting the possibilities to act as immune checkpoint inhibitors in future investigations.

## 4. Discussion

Since its characterization, *Penicillium* has become one of the most attractive fungal genera regarding the production of bioactive metabolites. However, Penicillium sp. derived from marine environments provides greater potential for lead compound discovery, ecology research, and environmental remediation, in light of the ecological diversity and metabolic plasticity for niche adaptation [24]. In this study, an endophytic *Penicillium* strain, B9, was derived from sponges in the South China Sea, and two novel pyridinone compounds were characterized from the fermentation products.

The strain was taxonomically identified as a distinct species in the genus of *Penicillium*, characterized by the high homology to the type strain ATCC 10480, in terms of the morphologic properties and phylogenetic analysis. The entire ITS region of B9 exhibited relatively high sequence similarity (97.8–98.9%) with the orthologues of *P. paxilli*. However, it was lower than the taxonomic threshold (99.6%) for fungal identification at the species level, but over the distinguishing genus threshold (94.3%) [25,26]. The case was reproduced in the phylogenetic classification by an additional DNA marker, calmodulin(CaM) (Appendix A).The phylogenetic classification of B9 was substantiated by the subtle variances of morphological features and ecophysiological adaptability, in comparison with *P. paxill* ATCC 10480, the type strain of *P. paxilli*.

As for this unique species, the whole genome was sequenced, with the aim of obtaining more information on genetic and functional content, which delineated a variety of BGCs. Albeit that, among the total 34 putative BGCs, only a few of them (4) shared ≥ 50% sequence similarity with annotated BGCs in the antiSMASH database; contrarily, uncharacterized gene clusters were predominant, which might confer an inconceivable synthetic capacity for the structural or biological diversification of natural products derived from the strain. Chemical separation partially verified the hypothesis, which characterized 3 kinds of SMs: penicipyridinone A, B, and paxilline.

Paxilline is a diterpene indole polycyclic alkaloid, abundantly produced by a cluster of 21 genes at the *PAX* locus in filamentous fungi, e.g., *P. paxillin* [23]. In terms of the stepwise biosynthetic process of paxilline, six *pax* genes (*pax*G-C-M-B-P-Q) were proposed to be necessary, involving two epoxidations, two cyclizations, two subsequent oxidation reactions, and a demethylation step [27]. The presumable *pax* cluster in the B9 genome exhibited extraordinary similarity in gene composition and content to the recorded counterparts, of which, the biosynthetic capacity was confirmed by the identification of the corresponding product.

Besides paxilline, two kinds of novel pyridinones, penicipyridinone A and B, were characterized. Intriguingly, various pyridinone-containing compounds were addressed to act as antagonists of programmed death protein 1 (PD-1) and the PD-L1 interaction [28,29], plausibly underpinning the functional relevance of this structure panel. Molecular docking and the subsequent HTRF assay indicated those compounds might be potential PD-1 pathway inhibitors for immune checkpoint blockage through the mechanism of PD-L1 dimer-locking.

## 5. Conclusions

The polyphasic approach used in this study, including morphological survey, comprehensive phylogenetic analysis, ecological investigation, and whole-genome sequencing, delineated that B9, a fungal strain derived from South China, should be a specific species in the genus *Penicillium*, sharing close affinity with *P. paxilli*. Investigation of the SMs from ferment extract led to the isolation of paxilline and two kinds of neo-pyridinones (penicipyridinone A and B). As for the latter, the absolute configurations were inferred from the NMR and ECD spectra analysis; whereas, the structure-based prediction and investigations of the compounds demonstrated as blocker to PD-1/PD-L1 interaction in vitro. The initial discoveries in our study threw some light on the ecological and genetic profiling of this newly discovered strain of *Penicillium* sp., which remains of great interest in further investigation into the biosynthetic potential and mechanisms of metabolites 

## Figures and Tables

**Figure 1 jof-08-00686-f001:**
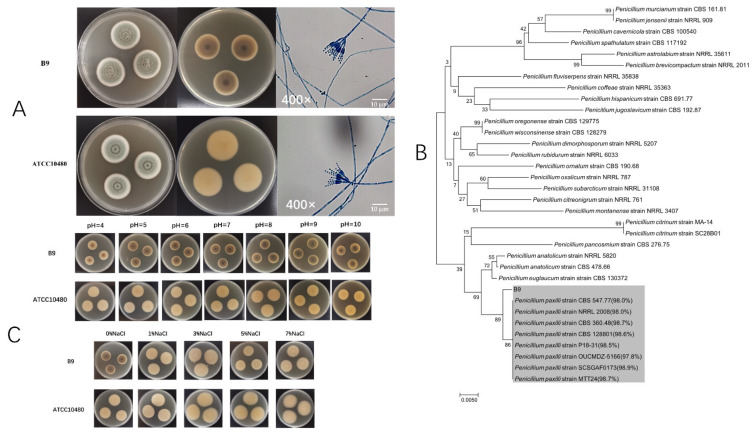
Analysis of species of the B9 strain. (**A**) Colony morphologies of Strain B9 and ATCC 10480 on MEA at 28 °C after 5 days. From left to right: obverse colonies on MEA, reverse on MEA, and observation results under an optical microscope of 400× magnification (scale bar 10 µm). (**B**) Phylogenetic tree of strain B9 based on the ITS sequence; the evolutionary history was inferred using the neighbor-joining method. Phylogenetic analyses were conducted with 1,000 bootstrap replications in MEGA7. Numbers above branches indicate bootstrap values. (**C**) Comparison of the physiological characteristics between B9 and *P. paxilli* ATCC 10480 under different pH values (upper) or salinities (lower).

**Figure 2 jof-08-00686-f002:**
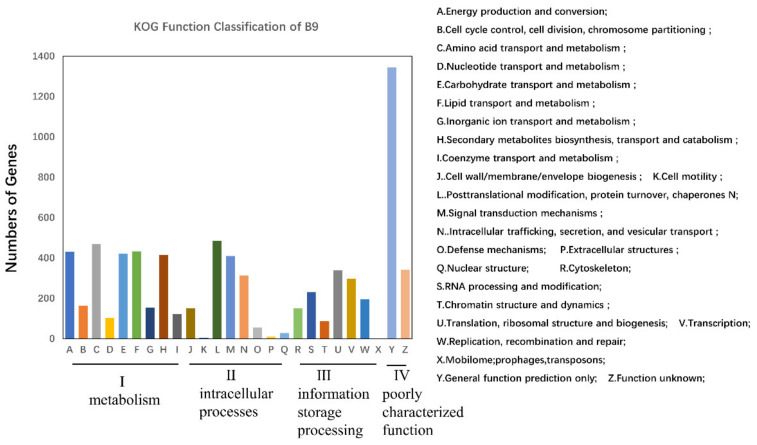
KOG classifications of putative proteins in the genome of B9. I: metabolism; II: intracellular processes; III: information storage/processing; IV: poorly characterized function.

**Figure 3 jof-08-00686-f003:**
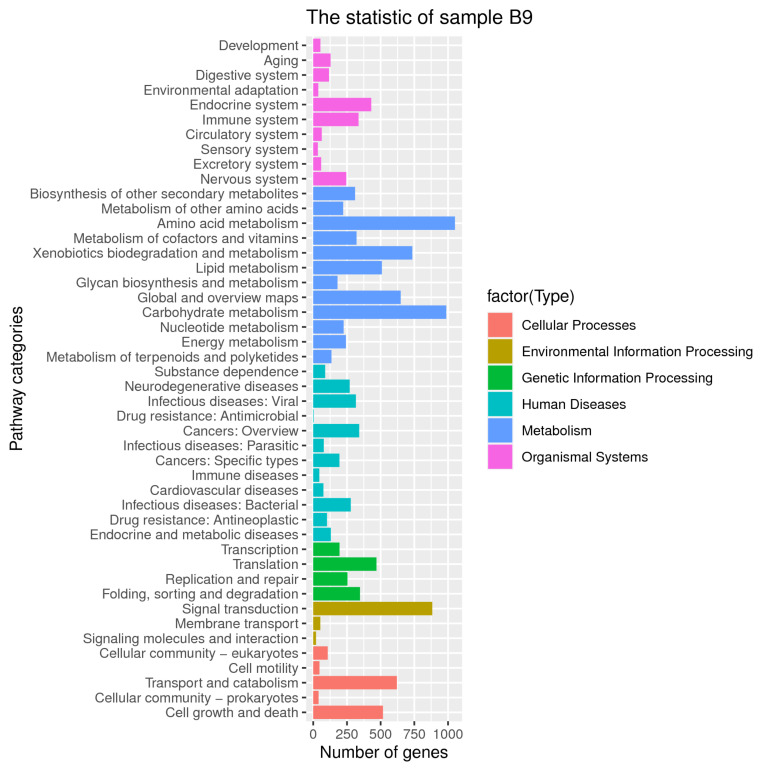
KEGG classifications of putative proteins in the genome of B9.

**Figure 4 jof-08-00686-f004:**
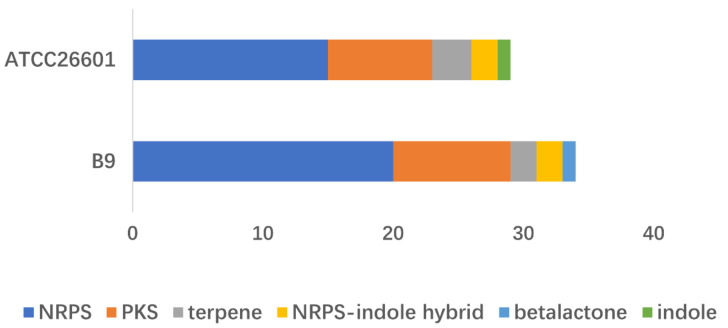
BGCs comparison between B9 and ATCC 26601. Different colors represent different BGC types.

**Figure 5 jof-08-00686-f005:**
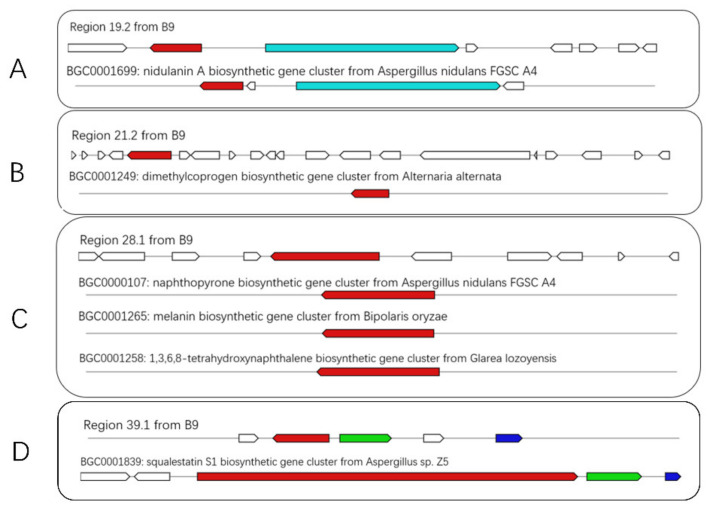
Schematic representation of B9 putative BGCs showing high homology with genes from characterized BGCs (**A**–**D**). The upper part represents the BGC in B9, and the lower part represents known BGCs in the MIBiG database. Homologous genes are shown in the same color.

**Figure 6 jof-08-00686-f006:**
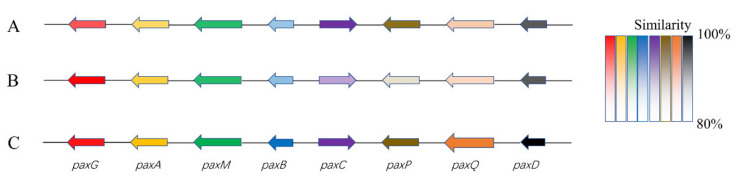
Schematic comparison of the putative BGC (pax) of paxilline found in B9 (**A**), reference genome (**B**), and literature (**C**). (**A**). The presumptive pax cluster in the B9 genome (B9 *pax*); (**B**). The presumptive pax cluster in reference genome (GCA_000347475) of *P. paxilli* ATCC 26601 (PP *pax*); (**C**). The defined *pax* gene cluster (BGC0001082; PAX) [23]. Different colors represent different genes, and the degree of similarities is distinguished by color shade.

**Figure 7 jof-08-00686-f007:**
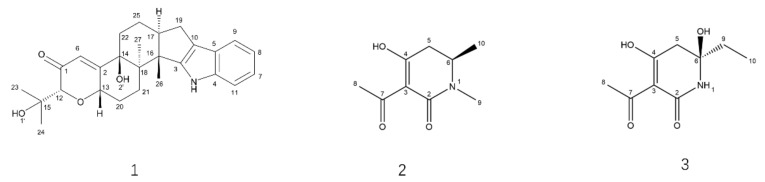
Structures of Compounds **1**–**3**.

**Figure 8 jof-08-00686-f008:**
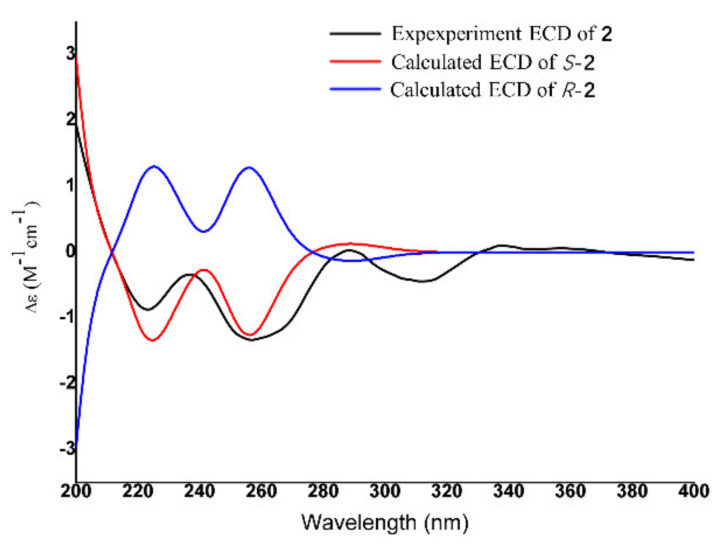
Experimental ECD spectra (200−400 nm) of 2S in methanol and the calculated ECD spectra of the model molecules of 2S at the PBE0/def2-TZVP level.

**Figure 9 jof-08-00686-f009:**
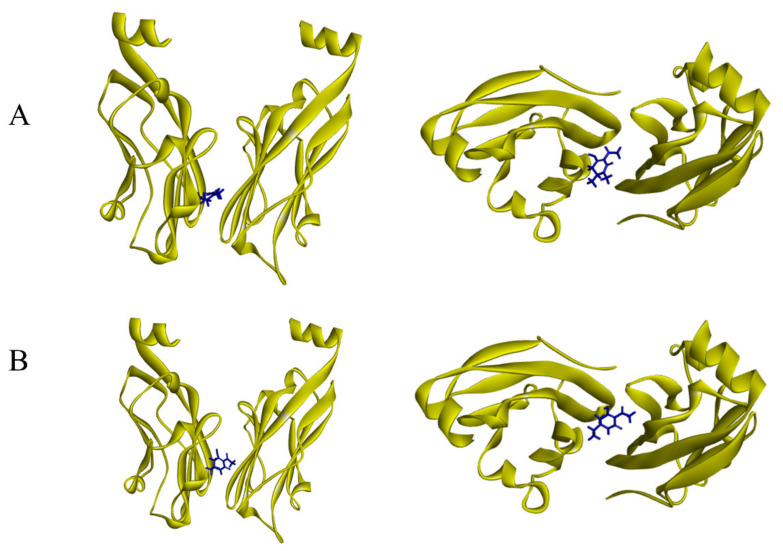
Results of docking simulation from different angles. (**A**) Binding modes of 2 at the PD-L1 dimer interface. (**B**) Binding modes of 3 at the PD-L1 dimer interface.

**Table 1 jof-08-00686-t001:** General features of the B9 genome.

Genome	Value
Assembly size (Mp)	32.96
G + C (%)	47.7
Assembled scaffolds	185
N50 length (bp)	1,131,010
average length (bp)	186,810
Predicted Protein-Coding Genes	11,110
average length of Predicted Protein-Coding Genes (bp)	1545.92
average depth of reads cover	189.27
total non-coding RNA	269
Sequencing Method	Illumina HiSeq

**Table 2 jof-08-00686-t002:** NMR data of compound **2** (500 MHz for ^1^H NMR, 125 MHz for ^13^C NMR, CD_3_OD).

No.	^13^C NMR	^1^H NMR	HMBC
1			
2	170.7		
3	109.7		
4	191.8		
5	42.8	2.49 (1H, dd, 15.9, 5.2); 2.34 (1H, dd, 15.9, 6.2)	C10, C6, C4, C3
6	47.0	3.76 (m)	C10, C5, C4, C2
7	198.9		
8	31.2	2.41 (3H, s)	C7
9	21.2	2.37 (3H, s)	C2
10	17.8	1.32 (3H, d, 6.6)	C6, C5, C4

**Table 3 jof-08-00686-t003:** NMR data of compound **3** (500 MHz for ^1^H NMR, 125 MHz for ^13^C NMR, CD_3_OD).

No.	^13^C NMR	^1^H NMR	HMBC
1			
2	179.7		
3	107.9		
4	201.6		
5	39.2	2.87 (1H, d, 15.2); 2.62 (1H, d, 15.2)	C4, C5, C6, C9
6	77.3		
7	197.1		
8	27.3	2.42 (3H, s)	C3, C7
9	30.1	1.66 (1H, dt, 17.5, 6,2); 1.73 (1H, dt, 17.5,6.2)	C4, C5, C6, C10
10	6.85	0.89 (3H, t, 6.2)	C6, C9

## Data Availability

Not applicable.

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
