# Peer review of "Genomic and Chemical Profiling of B9, a Unique Penicillium Fungus Derived from Sponge"

_jof, 2022, doi:10.3390/jof8070686_

Round 1

Reviewer 1 Report

Dear authors,

You have provided an account of an interesting isolate of marine fungi, Penicillium sp. B9, the sequencing of its genome containing many novel BGCs (30 out of 34 predicted. This could potentially generate several new chemical backbones with unusual activities to be investigated further for example as putative drugs leads. Described as a case in point are the structure of two chemically related and novel pyridinones compounds made by this strain and their inhibiting abilities in an in vitro model of programmed cell death protein 1 and programmed cell death 1 ligand 1.

Overall, I find this manuscript to be quite clear and well written and one that I enjoyed reading. However, I have some linguistic suggestions and points to consider as outlined below.

Line 41:  …flamentous fungal strains…; should read: …filamentous fungal strains…

Line 49: Here the introduction to why this in vitro model was chosen is missing.

Line 77-80: Seemingly, B9 might be an exceptional strain in Penicillium sp., represented by the different ecophysiological characteristics even from P. paxilli, its closest phylogenetic relatives, for which, further investigation was necessary. This sentence needs to be rephrased to become clearer.

Figure 1A. I cannot see the scale bar mentioned in line 84.

Figure 1B. The sequence of the type strain Penicillium paxilli, ATCC 10480 seems to missing in the phylogenetic tree.

Figure 1C. Line 88: …different pH values (upper) or salinities (down). Should read: … different pH values (upper) or salinities (lower). You could also write: … different pH values (top) or salinities (bottom). Please use the same suggested terminology throughout, for example in Figure 5 (Line 149).

Line 127: for clarity the full species name (Penicillium paxilli Bainier) should also be written out for when introducing the reference genome of ATCC 26601.

Line 156-159: The case should largely ascribe to the limited information on the structural and functional properties of the biosynthetic pathways available in the database, probably inadequate to support the comprehensive assessment on the strain, only from genome-mining scope. This sentence needs to be rephrased to become clearer

Line 228: …marine provide…, should read: …marine environments provide...¨

Line 233-234: The strain was taxonomically identified as a distinct species in the genus of P. paxilli,

characterized by the high homology to the type strain ATCC 10480. Needs to be rewritten. Penicillium is the Genus, P. paxilli is a species and ATCC10480 and B9 are strains. B9 could well be an entirely new species as supported by ITS sequencing, morphology etc. or perhaps a very odd strain of P. paxilli; the taxonomic nature of B9 is not yet fully determined. You could consider the possibility to provide the further details needed to propose this isolate B9 as a new species of Penicillium according to the below publication:

Aime, M.C., Miller, A.N., Aoki, T. et al. How to publish a new fungal species, or name, version 3.0. IMA Fungus 12, 11 (2021). https://doi.org/10.1186/s43008-021-00063-1

Final remark: in the Conclusion section, you argue that B9 should be regarded as a unique species. In that case, is not this one of your main findings? Should not this be spelled out more clearly in the Abstract?

Reviewer 2 Report

The work was done at a high technological level. Via the Illumina MiSeq sequencing platform, the draft genome of Penicillium strain B9 was sequenced and 34 BGCs were predicted, 4 of which the authors were able to match to previously known BGCs. The authors were also able to isolate two novel pyridinones, penicidihydropyridone A and penicidihydropyridone B, and show the potential pharmaceutical value of these compounds in silico. Elegant experiments were carried out, which allowed the author to obtain these novel natural products from this fungus.

The authors provide genome raw data sequencing of Penicillium strain B9 (PRJNA808833). For some reason online they titled bioproject for B9: Penicillium paxilli genome sequencing, not Penicillium sp. genome sequencing https://www.ncbi.nlm.nih.gov/bioproject/PRJNA808833 . Some confusion may arise from this, because in this article, the authors still do not dare to name Penicillium paxilli to B9, but online they decided. In order to find the nearest species in Penicillium, identification by ITS is not enough (it is also required to do a phylogenetic analysis for additional markers, such as b-tubulin (BenA), calmodulin (CaM), or RNA polymerase II second largest subunit (RPB2) genes. [Page 349 - Species Diversity in Aspergillus, Penicillium and Talaromyces | NHBS Academic & Professional Books Available online: https://www.nhbs.com/species-diversity-in-aspergillus-penicillium-and-talaromyces-book]. Judging by how much B9 differs from Penicillium paxilli in terms of BGCs (Figure 4.), these are different species.

The authors also describe in detail how they annotated the genome and say that Predicted 11,110 Protein-Coding Genes, as well as 34 biosynthetic gene clusters (BGCs), but did not post these data either in the supplementary, but online. Post the genome annotation data and BGCs prediction data online. Otherwise, the data on their number, which you provide in the article, are unfounded. Also give information on the number of contigs and  scaffolds you have docked.

Comments

Line 33

Correct indol-diterepene to indol-diterpene

Line 42-45

Sequence the regions needed to identify your Penicillium: (b-tubulin (BenA), calmodulin (CaM), or RNA polymerase II second largest subunit (RPB2). This will allow you to more specifically select the reference strain. Perhaps it will remain P. paxilli, or perhaps you will find a closer species described in the literature and studied.

To determine the fungi genus Penicillium to species, genotyping for the ITS is not enough. To accurately determine the species, you need to use b-tubulin (BenA), calmodulin (CaM), or RNA polymerase II second largest subunit (RPB2) genes. [Page 349 - Species Diversity in Aspergillus, Penicillium and Talaromyces | NHBS Academic & Professional Books Available online: https://www.nhbs.com/species-diversity-in-aspergillus-penicillium-and-talaromyces-book].

Line 82

 Reformulate the common name figure from “Results of strain identification”. In the above experiments, you did not identify the strain, but demonstrate that, on the contrary, it differs from the comparable strains described. Perhaps you have a new strain that will eventually be given a specific name.

 Figure 1A

The caption to the figure mentions the scale bar, but the scale bar is missing on the photos themselves. Add scale bar

 Figure 1C

Arrange more carefully the pH and NaCl signatures at equal distances from the photos

 Line 161

Reword the title of the picture. Make it more specific.

Figure 6.

Give in the caption to figure 6 (for subparagraphs A, B, C) the names of the organisms in which paxilline BGC was compared

 Line 161-163

Make text in one font

Lines 309-313

Post the genome annotation data and BGCs prediction data online. Otherwise, the data on their number, which you provide in the article, are unfounded.

 Line 318

“8 kg fermented substrate”

How much did you ferment to get 8 kg of fermented substrate? Write in the article how much you used rice medium.

 Lines 306-307, 364-365

“fungal strain derived from South China, should be a specific species in genus Penicillium, sharing close affinity with P. paxilli.” (c)

 In this article, you do not directly say that Penicillium strain B9 is directly P. paxilli. Then why did you annotate the genome of this B9 strain as P. paxilli? https://www.ncbi.nlm.nih.gov/sra/?term=PRJNA808833

 Design.

Arrange the references in accordance with the requirement of the journal (put the reference numbers in square brackets, not superscript).

Abbreviations

Give transcripts abbreviation uniformity. Example: MEA - Malt Extract Agar, PDA - potato dextrose agar. In one case, the decoding of the name of the environment is in capital letters, in the other, in lowercase.

 References

 Line 381 and beyond

Latin names - italicize
